spectroscopy/green chemistry/plant science

rosehip hypanthium, rosehip seed, Raman spectroscopy, PCA, carotenoids, polyphenols

**Author for correspondence:**
Jelena B. Popović-Djordjević
e-mail: jelenadj@agrif.bg.ac.rs

This article has been edited by the Royal Society of Chemistry, including the commissioning, peer review process and editorial aspects up to the point of acceptance.

[†]Present address: Department for Chemistry and Biochemistry, Faculty of Agriculture, University of Belgrade, Nemanjina 6, 11080 Belgrade, Serbia.

# Rapid characterization of hypanthium and seed in wild and cultivated rosehip: application of Raman microscopy combined with multivariate analysis

Ilinka Pećinar[1], Djurdja Krstić[2], Gianluca Caruso[3] and Jelena B. Popović-Djordjević[4,†]

[1]Faculty of Agriculture, Department for Agrobotany, University of Belgrade, Nemanjina 6, 11080 Belgrade, Serbia
[2]Faculty of Chemistry, University of Belgrade, Studentski trg 12-16, 11158 Belgrade, Serbia
[3]Department of Agricultural Sciences, University of Naples Federico II, Portici (Naples), Italy
[4]Faculty of Agriculture, Department for Chemistry and Biochemistry, University of Belgrade, Nemanjina 6, 11080 Belgrade, Serbia

JBP-Dj, 0000-0003-4057-3826

Rosehip (pseudo-fruit) of dog rose (*Rosa canina* L.) is highly valued, and owing to nutritional and sensory properties it has a significant place in the food industry. This work represents an innovative report focusing on the evaluation of the phytochemical composition of rosehips (hypanthium and seed) grown in different locations in Serbia, using Raman microspectroscopy combined with multivariate data analysis. Some significant differences arose between the analysed rosehip samples with regard to the chemical profile of both hypanthium parenchyma cells and seed, although no evident discrimination was recorded between the samples of wild and cultivated rosehip. The differences between the hypanthium samples compared were mainly determined by the content of carotenoids, phenolic compounds and polysaccharides, whereas phenolics, polysaccharides (pectin, cellulose and hemicellulose) and lipids (to a lower extent) contributed to the seed sample discrimination. The differences observed between the rosehip samples may be attributed to abiotic factors (growing, ripening and storage conditions), which had a significant impact on the carotenoid and polyphenols biosynthesis.

# 1. Introduction

Dog rose (*Rosa canina* L.) is a wild shrub growing in temperate to subtropical habitats of Europe, western Asia, Middle East, North America and the northwest part of Africa. This well-known species has been traditionally used and recently considered as a complex of species (the aggregate), owing to its genetic variability and related morphological polymorphism resulting from the interaction between the environmental conditions and the genetic background. Several studies have shown the close relationship between the genetic and phytochemical diversity of different *Rosa* species [1–3].

Dog rose has been traditionally used as a source of many bioactive molecules mainly contained in its pseudo-fruit (rosehip) but also in petals and leaves. The ancient Romans believed that this plant was even effective against rabies and, therefore, they used *R. canina* (from the Latin word canis, meaning 'dog') in cases of rabid dog bites, although no scientific association has been found in the literature. The rosehip of this species has many beneficial effects, such as anti-diarrhoeic, astringent, healing, depurative, diuretic and venotonic, and it is mainly used as infused, tinctured or consumed as syrup or dragees. On the other hand, the seeds are a rich source of polyunsaturated fatty acids and were used in folk medicine in wound-healing and skin disease treatments [4]. Rosehips and/or petals can be used in gastronomy (e.g. salads, confiture, honey, herbal tea, vine, liqueur, etc.), food industry (marmalade, jam products, probiotic beverages, fruit yogurts and soup production) and cosmetic industry (rosewater, anti-ageing oil, anti-acne and pigmentation-reducing formulations) [4,5,6].

*Rosa canina* has been broadly studied for its phytochemical, nutritive and wide range of medicinal properties. Rosehips have been found to contain higher amounts of various bioactive compounds than several other fruits and berries [7]. The fruits of different rose species are a rich source of carotenoids, proteins, sugars, organic acids, phenolic compounds, essential elements and vitamins (B-group, C and E) [5,6,8–12]. The health benefits of rosehips are mainly linked to the presence of bioactive compounds, such as a high content of carotenoids, mainly represented by lycopene and β-carotene, and only traces of lutein, zeaxanthin and rubixanthin [13], in addition to the remarkable content of ascorbic acid, total polyphenols, tocopherol, pectin and other metabolites. Eight individual phenolic compounds such as a apigenin, cinnamic acid, quercetin, rutin, *p*-coumaric acid, chlorogenic acid, caffeic acid and gallic acid were identified in the studied rosehips [3]. In rosehip seeds, carbohydrates and proteins were found to be the most and the least abundant primary metabolites, respectively [14], and the lipid fraction contains more than 50% polyunsaturated fatty acids [15]. Linoleic acid, linolenic acid and oleic acid are the major fatty acids in the rosehip [14], followed by palmitic, stearic and arachidic acids at lower concentration [16]. The high ratio of linolenic acid to linoleic acid may make rosehip seed oil a valuable source of omega fatty acids. All the constituents of rosehip exhibit various health-promoting effects, such as reducing oxidative stress, preventing heart disease, cancer, osteoarthritis and skin protection [5,17].

The following techniques have been mostly used for analysing qualitatively and quantitatively the rosehip bioactive compounds: high-performance liquid chromatography with diode array detection for phenolics and flavonoids [3,18]; thin-layer chromatography and high-performance liquid chromatography for carotenoids [7,13,18–20]; and gas chromatography with flame-ionization detection for fatty acids [16,21,22].

In the past, Raman spectroscopy (RS) was primarily used for research, but nowadays it is being widely used for providing data relevant to chemical composition and structural characteristics (so-called fingerprint) as well as for semi-quantitative and quantitative analyses of various plant materials [23,24]. The *In situ* analysis by RS is a rapid and non-destructive method requiring minimal sample pre-processing and therefore can be considered an eco-friendly assay. In the latter respect, RS entails some advantages for the rose fruit and seed characterization when compared with the abovementioned analytical methods. Although many plant species have been analysed by RS the only application of this method for a successful characterization of rosehip (*Rosa aff. rubiginosa*) was reported by da Silva *et al.* [25]. The combination of RS results and appropriate multivariate analysis includes the simultaneous determination of different parameters and the quantification of some other variables, which is interesting for the characterization of the target samples. The statistical analysis of the recorded spectra provides the possibility to analyse the data containing superposed signals and the simultaneous determination of several constituents [26].

The rosehip of dog rose is highly appreciated throughout the World, especially in many European Countries, owing to its nutritional value and sensory properties, and has a significant place in the food industry. A previous study highlighted that the Serbian rosehip is a valuable source of biologically active compounds [2]. In addition, seeds from both cultivated and wild plants are rich in

**Table 1.** Samples labels and areas of sampling.

| sample | | | sample | | |
| --- | --- | --- | --- | --- | --- |
| hypanthium (H) | seed (S) | | hypanthium (H) | seed (S) | |
| wild rosehip (W) | | sampling area | cultivated rosehip (C) | | sampling area |
| 1 HW | 1 SW | Lazarevac (Rudovci) | 7HC[a] | 7 SC[a] | Nova Varoš (Seništa) |
| 2 HW | 2 SW | Kraljevo (Gokčanica) | 8 HC[a] | 8 SC[a] | Valjevo (Mrčić) |
| 3 HW | 3 SW | Mionica (Sankovići) | 9 HC[b] | 9 SC[b] | Nova Varoš (Seništa) |
| 4 HW | 4 SW | Kraljevo (Samaila) | 10 HC[b] | 10 SC[b] | Valjevo (Mrčić) |
| 5 HW | 5 SW | Obrenovac (Mala Moštanica) | | | |
| 6 HW | 6 SW | Čačak (Beljina) | | | |

[a]'Laksa' cultivar.
[b]'Polimerijana' cultivar.

nutritionally valuable elements [11]. However, the available data regarding the phytochemical and nutritional composition of rosehip, especially its seeds, are still very limited.

Aiming to continue our research related to Serbian rosehip, the purposes of the current study were: (i) to evaluate the phytochemical and nutritional composition of the cultivated and wild rosehips collected from different locations in Serbia using Raman vibrational characterization; (ii) to assess the validity of a method for rapid sample screening; and (iii) to differentiate the samples based on their chemical composition through the application of principal component analysis (PCA) of the obtained Raman spectra. From an accurate literature survey, it has arisen that this, to our knowledge, is the first report regarding the chemical composition evaluation of *Rosa* fruits grown in Serbia using Raman microspectroscopy.

# 2. Material and methods

## 2.1. Rosehip (hypanthium and seed) material

Rosehip (*R. canina* L.) samples were taken at eight locations in Serbia (table 1). Wild fruit samples were grown at six different locations, whereas two cultivars named 'Laksa' and 'Polimerijana' were grown in two orchards located in Valjevo and Nova Varoš regions [11]. Wild and cultivated mature rosehips were manually harvested in September–October 2018, and samples consisting of 50 fruits were randomly taken from different bushes at each location. In order to perform the RS analysis, the seeds were separated from hypanthium in each sample, placed in plastic boxes and kept at −18°C until measuring. Further, just before recording spectra, the seeds were longitudinally cut at room temperature, whereas the fleshy fruit parts (hypanthium) were homogenized. The plant herbarium materials were deposited in the Herbarium of the Faculty of Agriculture in Belgrade (Zemun), Serbia.

## 2.2. Raman instrumentation

Raman microspectroscopy of rosehip fruit focused on the direct measurement of rosehip hypanthium parenchyma cells and its seeds. Spectra were recorded using XploRA Raman spectrometer (Horiba Jobin Yvon). With regard to seeds, Raman scattering was excited by laser at a wavelength of 785 nm equipped with a 600 lines $mm^{-1}$ grating; spectra were recorded by applying exposure time 20 s and accumulated from 20 times scans, using 100% filter. Raman spectra for hypanthium parenchyma cells were recorded using a laser at a wavelength of 532 nm equipped with a 1200 lines $mm^{-1}$ grating; spectra were acquired by applying exposure time 10 s and scanning the sample 10 times, using 10% filter. The spectral resolution was about 3 $cm^{-1}$ and autocalibration was done each time before recording of spectra by 520.47 $cm^{-1}$ line of silicon. In order to assess a possible sample inhomogeneity, 10 Raman spectra were recorded in each sample. The identification of the major bands was carried out using literature data (see the electronic supplementary material, table S1).

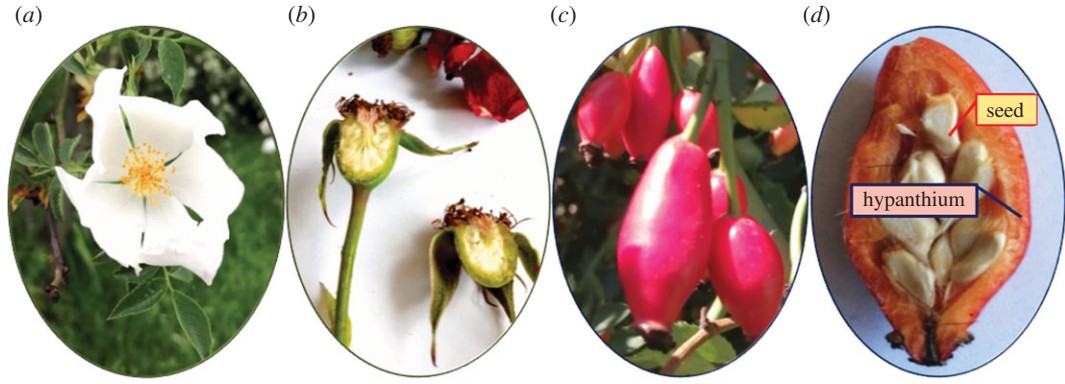

**Figure 1.** (*a*) Flower of *Rosa* sp., (*b*) flower longitudinal section: multiple ovaries are positioned on the side of a urn-like structure known as the hypanthium, (*c*) aggregate fruits-rosehip consisting of, (*d*) several achenes (one seed-containing fruits) surrounded by thin pericarp, enlarged by fleshy floral cup (hypanthium). (Photos taken by J. Popović-Djordjević in Čačak sampling area).

## 2.3. Statistical analysis

The PCA was performed by means of PLS TOOLBOX, v. 6.2.1, for Matlab 7.12.0 (R2011a). The PCA was carried out as an exploratory data analysis by using a singular value decomposition algorithm and 0.95 confidence level for Q and T2 Hotelling limits for the outliers. The PCA was applied to the data obtained from Raman spectra at the range 200–1800 cm$^{-1}$ for seed and hypanthium samples, respectively. Prior to multivariate analysis, the raw data were pre-processed using the SPECTRAGRYPH software [27] baseline correction (weighted least squares) to reduce the baseline drift influence, followed by normalization and mean centering procedures. The standard normal variate, as a commonly applied normalization technique, normalizes the variables from each spectrum to zero and scales them by the standard deviation of the spectral data. On the other side, mean centering is the preferred option when the sample classification is based on parameters measured by the same unit.

# 3. Results and discussion

## 3.1. Rosehip botanical description

The pseudo-fruits (rosehip) of *R. canina* (figure 1), often referred to as 'fruits' in literature sources, are aggregate fruits consisting of several achenes (the actual one seed-containing fruits of dog rose) enclosed by an enlarged fleshy floral cup (hypanthium) [28]. The rosehip has been analysed at macro- and micro-scale through achene morphology [29,30], morpho-anatomical characterization of achenes, mainly pericarp [31,32,33] and seed morphology [34]. In the upper part of the hypanthium, there is a small opening, filled with protruding styles (figure 1*b*). During the rosehip development, the walls of the flower hypanthium become fleshy and red (figure 1*c,d*), while the apocarpous gynoecium turns into many achenes [30,32]. Inside the hypanthium, the thin fruit membranes (pericarp) surround the individual achenes (figure 1*d*) forming an aggregate fruit, altogether called a 'pseudo-fruit' of dog rose plants [32].

## 3.2. Raman 'signature' of hypanthium and seed reserves

The bands in Raman spectra correspond to specific bands of chemical bonds and/or functional groups of the molecule, and the band intensity is linked to the quantity of the analysed compound. The specific bands make the so-called 'fingerprint' of the molecule. On the other hand, the peculiarity of Raman bands lies in the fact that their intensity does not always arise from the principal components in the complex matrix, which depends on the molecular scattering behaviour of that chemical species [25,35]. Some common chemical characteristics of polyenes, despite of the variety of their molecular structures, make these compounds very suitable systems for RS analysis [36]. The analysis of the fruit compartments (hypanthium and seed) was obtained by Raman microspectroscopy, and the averages

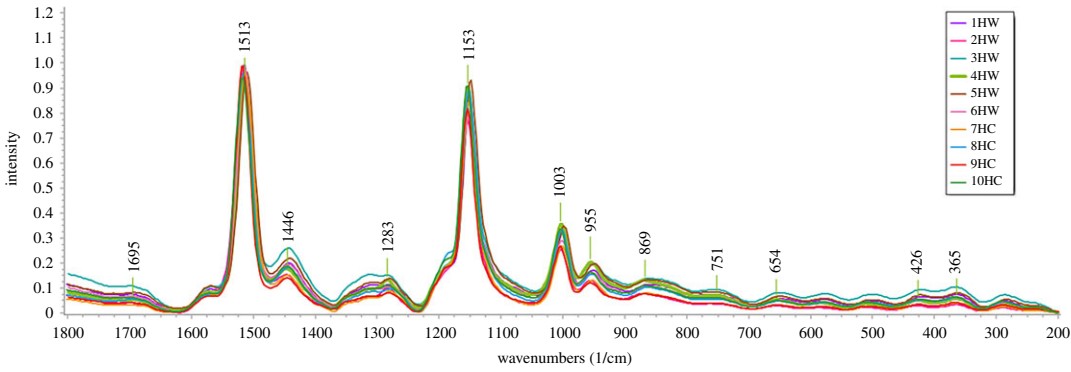

**Figure 2.** Averages of normalized Raman spectra of wild (1–5 HW) and cultivated (6–10 HC) rosehips recorded from hypanthium parenchyma cells in the spectral range from 200 to 1800 cm$^{-1}$.

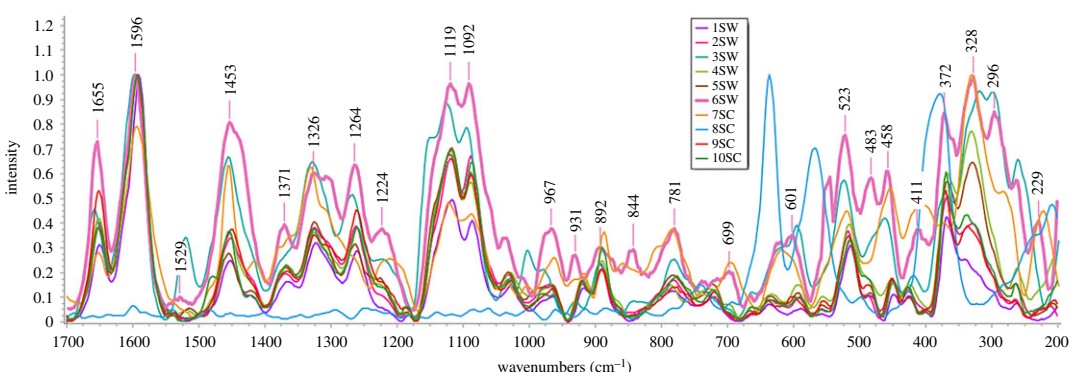

**Figure 3.** Averages of normalized Raman spectra of seeds in wild (1–5 SW) and cultivated (6–10 SC) rosehip collected from different regions of the Republic of Serbia in the spectral range from 200 to 1800 cm$^{-1}$.

of hypanthium and seed spectra related to wild and cultivated rosehip samples (table 1) are presented in figures 2 and 3, respectively. The characteristic vibrational bands and the related tentative identification within the spectra acquired from tissue sections of the studied rosehip hypanthium and seed samples are shown in the electronic supplementary material, table S1.

*Hypanthium.* In the Raman spectra obtained from wild and cultivated rosehip hypanthium samples (1H–10H), significant bands associated with some hypanthium essential constituents were observed at 1513 cm$^{-1}$ (very strong), 1153 cm$^{-1}$ (very strong) and 1003 cm$^{-1}$ (medium) (figure 2). These are the main characteristic bands of carotenoids ('carotenoids fingerprint') and can be assigned as the stretching of the C = C (*v*1), C−C (*v*2) bonds and the C−CH$_3$ in-plane group rocking vibrations ($\rho$(C−CH$_3$)), respectively. This suggests the presence of carotenoids, the major constituents of rosehip parenchyma cell (main tissue of hypanthium), consistently with the reports of the only study related to *Rosa* sp. fruit (*Rosa aff. rubiginosa*) (see the electronic supplementary material, table S1), where the bands relevant to carotene were at 1520 cm$^{-1}$ (very strong), 1157 cm$^{-1}$ (medium strong) and 1007 cm$^{-1}$ (weak) [25]. Literature data regarding different plant samples also associate these three bands with the presence of various carotenoids [36–38].

As shown in the Raman spectra of individual samples (electronic supplementary material, figure S1), the bands of the 'carotenoids fingerprint' are at different wavelengths, which suggests differences between the samples in terms of the qualitative and quantitative composition of carotenoids. However, assigning the spectral data of 'carotenoids fingerprint' to a specific compound is not easy, as the carotenoids may interact with other compounds present in plant tissue [38,39]. Very weak bands were identified at 1695 cm$^{-1}$, and in the region from 870 to 280 cm$^{-1}$, and weak bands at 1446, 1283 and 955 cm$^{-1}$ (figure 2). In the literature, the amide I bands, very useful for elucidating the protein secondary structure, were observed at 1600–1700 cm$^{-1}$ [35,40]. In the latter respect, the vibrational band at 1695 cm$^{-1}$ could be tentatively assigned to proteins (figure 2; electronic supplementary material, table S1).

According to da Silva *et al.* [25], the band at 1446 cm$^{-1}$ is related to the $\delta(CH_2)$ vibrational mode and is associated with lipids and/or glucosidic structure in rosehip. Moreover, it was reported that linoleic acid ($v$(C-C)) and carotenoids (CH$_2$-scissoring vibration) provide a band in Raman spectra at 1440 cm$^{-1}$ [35]. Having in mind that the rosehip fruit is a good source of carotenoids [12] and unsaturated fatty acids such as linolenic, linoleic and oleic acid [41], the band 1446 cm$^{-1}$ could be associated with the lipid class. As rosehip flesh is rich in various carbohydrates [10,12], the bands observed below 1000 cm$^{-1}$ may probably be related to this class of compounds. Synytsya *et al.* [42] assigned the bands at 953, 365 and 751 cm$^{-1}$ to polygalacturonic (pectic) acid and pectin, respectively. The bands ranging from 280 to 869 cm$^{-1}$ probably refer to glucosidic link stretches [25], such as those at 290, 426 and 505 cm$^{-1}$ (figure 2, electronic supplementary material, table S1).

*Seed*. The average Raman spectra of the rosehip seed samples of the so-called fingerprint region (200–1800 cm$^{-1}$) are presented in figure 3, while characteristic bands and the related assignments are reported in the electronic supplementary material, table S1. The obtained spectra include bands that may be associated with the most important compounds found in rosehip seeds such as fatty acids [14,25,43], proteins [14,41], phenolic compounds and carbohydrates [10,14,25].

Phenolic compounds are constituents of rosehip seed that contribute to its antioxidant properties [10,14]. The highest intensity band (1596 cm$^{-1}$) in the average Raman spectra of seed samples (1S–10S), is correlated to $v$(C = C), the structural feature of phenolic compounds [25,44,45].

The bands related to carbohydrates, the most abundant macronutrients in the rosehip seed [14], appear with a higher intensity in the 1080–1460 cm$^{-1}$ region [25,45], and lower intensity in the region 400–800 cm$^{-1}$ [45,46]. Literature data (electronic supplementary material, table S1) indicate that pectin and its structural components have bands between 1453 and 372 cm$^{-1}$ [42,46]. Among the carbohydrates found in rosehip seed, fructose, glucose, sucrose and maltose were measured in different concentrations [12]. The tentative assignments of the bands recorded in seed samples 1S–10S (figure 3; electronic supplementary material, table S1) at wavelengths: 1453, 1371, 1326, 1264, 1037 cm$^{-1}$, and below 1000 cm$^{-1}$ could be associated with mono-, di- and polysaccharides [25,35,42,45–48].

Literature reports suggest that the unsaturated fatty acids (linolenic, linoleic and oleic) are the main fatty acids detected in the rosehip seeds, with the highest percentage of linoleic acid [14], while the saturated fatty acids (palmitic and stearic) show lower concentrations [41]. The Raman spectra of linolenic, linoleic and oleic acids exhibit two specific medium-intensity bands at 1655 and 1264 cm$^{-1}$ (see the electronic supplementary material, table S1) related to the presence of *cis* stretching vibration of C=C in the propanoic chain and the bending of C–H from unsaturated fats, respectively [25,37,49]. Additionally, the regions from 1654 to 1660 cm$^{-1}$ and from 1600 to 1700 cm$^{-1}$ are also associated with proteins, assigned to the amide I band (predominantly β-sheet) (electronic supplementary material, table S1) [35,40,50].

The complexity of the studied plant material was expected. The similarities and differences between the samples related to the composition and structural diversity of the detected compounds could be observed in the spectra of individual hypanthium and seeds (electronic supplementary material, figures S1 and S2, respectively). The biological samples commonly show multiple overlapping bands associated with a large number of chemical classes (each proteins, nucleic acids and lipids), which makes it difficult to match a certain band to a specific compound [38,49].

Sometimes individual Raman spectra do not show any distinctive visual differences between the samples compared. Hence, in this work PCA was used for highlighting the differences between the samples which are not evident in the mean spectra corresponding to the regions of rosehip hypanthium and seed (figures 2 and 3, respectively).

## 3.3. Principal component analysis of the data obtained from Raman spectra of rosehip hypanthium parenchyma cells

PCA has been performed to visualize the data structure and identify the influential variables. PCA was applied to the data obtained from Raman spectra in the range 200–1800 cm$^{-1}$, in order to get the criteria for separating the cultivated from wild rosehips (10 seed and 10 hypanthium) samples. Prior to multivariate analysis, all data were pre-processed as has been explained in §2. The first PCA model obtained for hypanthium samples resulted in three principal components explaining 95.32% of the total data variance. The first principal component (PC1) accounted for 83.86% of the overall data variance, while the second one (PC2) for 7.94%. Mutual projections of factor scores and their loadings for the first two PCs are shown in figure 4.

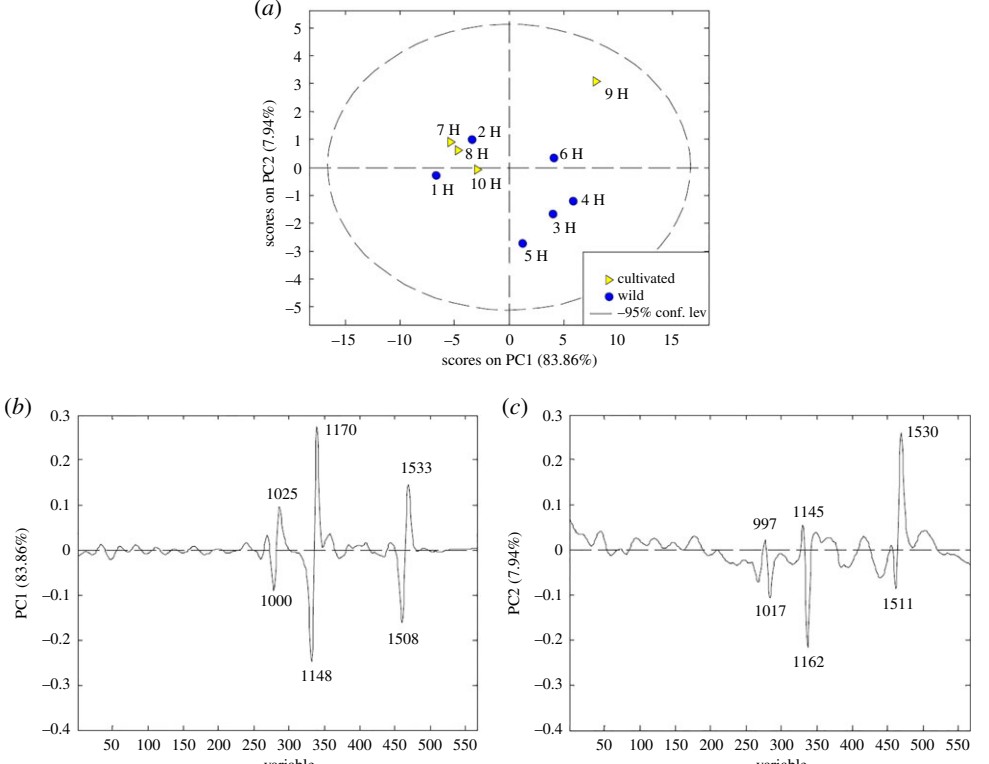

**Figure 4.** PCA analysis applied to the data obtained from Raman spectra of wild and cultivated rosehip hypanthium samples: (a) score plot, (b,c) loading plots.

The score plot (figure 4a) suggests the existence of two groups of objects along the PC1 axis. The first one includes hypanthium samples 3, 4, 5 and 6 HW from the wild and 9 HC, from cultivated rosehip, respectively. Samples 1, 2 HW and 7, 8 and 10 HC form the second cluster. The corresponding loading plot represents the relationship between the variables and can be used to identify those with the highest contribution to the object positioning in the score plot. The loading plot (figure 4b) shows that the variables with the highest positive contribution along the PC1 axis corresponded to the signals at 1025, 1170 and 1533 cm$^{-1}$, while signals at 1000, 1148 and 1508 cm$^{-1}$ have the highest negative effects. These variables are responsible for the differences between 7, 8, 10 (HC), 1 and 2 (HW) samples, and the wild rosehip samples 3, 4, 5 and 6 (HW) and 10 HC. The signal at 1170 cm$^{-1}$ is mostly responsible for the differentiation among the samples involving vibration of C=C and C–C vibration from the polyenic chain, together with the medium-intensity loading at 1533 cm$^{-1}$, attributed to carotenoids, respectively [36,37,51]. The third and lowest intensity signal at 1025 cm$^{-1}$ can probably be assigned to $\nu$(CC) and (CO) from polygalacturonic (pectic) acid [42]. The highest negative intensity loading placed at the position 1148 cm$^{-1}$ is probably assigned to carotenoids [25]; medium-intensity loading at 1508 cm$^{-1}$ might be associated with stretching vibration in carotenoids [35], the band at 1000 cm$^{-1}$ which is the third and the lowest negative signal specific of $\nu$(C–C) points to carotenoids [25].

The variables that potentially had the highest positive influence on the separation along PC2 (figure 4c) corresponded to the signals at 997, 1145 and 1530 cm$^{-1}$, whereas the variables at 1017, 1162 and 1511 cm$^{-1}$ had the highest negative effect. The differences between the samples 1, 3, 4, 5 (HW) and 10 HC compared to 2, 6 (HW) and 7, 8, 9 (HC) rosehips in the hypanthium region mainly depend on carotenoids, which indicates a high-intensity positive signal at 1530 cm$^{-1}$ [45,50], while a lower intensity loading at 997 cm$^{-1}$ points out the occurrence of cellulose or hemicellulose [42,52]. A lower intensity signal at 1145 cm$^{-1}$ could be assigned to the C-C stretching mode of conjugated polyenes [25]. The chemical profiles of 2, 6 HW, 7,8,9 HC differ from 1, 3, 4, 5 HW and 10 HC in respect to variables placed at the following positions: 1511 cm$^{-1}$, assigned to C=C vibration in carotenoids [44,53], 1162 cm$^{-1}$ which might be related to the C–O and C–CH vibration in pectin molecules [54], and the 1017 cm$^{-1}$ band which is the third negative signal, associated with phenolic compounds [55].

Although rosehip is generally known to contain a high amount of health-promoting compounds such as carotenoids and phenolic compounds, differences between rosehips placed in the hypanthium region may arise owing to numerous abiotic factors (climate, i.e. exposure to sunlight, temperature, land exposition, elevation, etc.) as well as ripening degree, growing and storage conditions. All the aforementioned conditions, together with biochemical traits, may affect the quantitative variation of pigments in rose fruits [3,7] because of their effects on carotenoids biosynthesis. Taking into account the various growing locations of the rosehips collected, it could be assumed that the different abiotic factors influenced carotenoids biosynthesis in most of the cultivated rosehips (7, and 8, 10 HC) as well as in wild rosehips (3, 4, 5, 6 HW), compared to all other locations (indicated by the positive loading of PC1 and PC2). According to the literature, no differences in β-carotene content were observed between some wild and cultivated rosehip [56]. On the other hand, a strong antioxidant activity was reported in wild *R. canina*, depending on the content of β-carotene in fruits [57] or polyphenols in different genotypes [58]. In agreement with the latter finding, the positive variable loading at 1017 cm$^{-1}$ on PC2 (figure 4c) shows that most of the wild rosehips (1, 3, 4, 5 HW) could be differing from the cultivated in phenolic content.

## 3.4. Principal component analysis of the data obtained from Raman spectra of rosehip seeds

Raman spectra provide information connected with the overlapping signals assigned to various plant components, especially in seeds with different composition of fatty acids, phenolics and carbohydrates. With the aim of extracting the latter information, the PCA analysis of the Raman spectra related to the seeds was performed. The second PCA resulted in a three-component model explaining 94.56% of the total data variance, whereas the first two PCs explained 90.64% of the total data variability. The mutual projections of factor scores and their loadings relevant to the first and second PCs are shown in figure 5.

The score plot (figure 5a) suggests the existence of two distinctive clusters along with the PC2 and PC1 directions. The first cluster consisted of 1, 2, 5 SW and 9, 10 SC samples at the lower left side of the plot, while the samples 3, 4, 6 SW and 7 SC formed the second cluster at the upper right side of the plot. This means that there is no clear differentiation between the cultivated and wild fruit seeds. Sample 8 SC showed a different profile compared to all other seed samples. The loading plot (figure 5b) revealed the variables which are mainly responsible for sample differentiation along the PC1 axis, with signals at 390, 576 and 637 cm$^{-1}$ (positive influence) and 1123, 1326 and 1601 cm$^{-1}$ (highest negative contribution) (figure 5b). The spectral range from 200 to 650 cm$^{-1}$ had the highest impact on the differentiation of the 8 SC sample; this spectral region involves the deformation modes of the C-C-O bonds at 575 and 637 cm$^{-1}$ as well as the C-C-C skeletal vibrations at 390 cm$^{-1}$, both the glycosidic ring skeletal deformations [26,52] indicating the predominance of cellulosic glycosides in the seed structure. Differently, the variables at 1123, 1326 and 1601 cm$^{-1}$ derived from cellulose and phenolic compounds, and affected the separation of 1, 2, 5 SW as well as the 9 and 10 SC samples. The signal at 1123 cm$^{-1}$, assigned to the symmetric stretching mode of C–O, C–C and C–O–H [25,26,46] is characteristic of the glycosidic bond vibration in cellulose or hemicellulose. It was found that the negative medium-intensity signal at 1326 cm$^{-1}$ is characteristic of the external seed parts and originates from cellulose [25], probably eliciting the C-O and C-O-H stretching vibrations [26]. The most important differences between the samples coming from the highest negative intensity signal at 1601 cm$^{-1}$, assigned to the aromatic C=C vibration of phenolic polymers, emphasized the dominance of the lignified tissue in the external seed structure [58]. The lignified tissue which is the most typical of the seed sclerenchyma fibres, is associated with a lower negative intensity signal (assigned to CH$_3$ bending). The phenolic acids (e.g. *p*-coumaric and/or ferulic acid) have been already described as the predominant phenolic compounds in the rosehip seed coat [59,60]. Among polyphenolic substances, lignin is the most representative, because it fills the spaces in the cell wall between cellulose, hemicellulose and pectin [25].

The most influential parameters along the PC2 axis corresponded to the signals at 312, 473 and 856 cm$^{-1}$ (positive impact) and signals at 579, 646, 1128, 1267 and 1599 cm$^{-1}$ with the highest negative impact (figure 5c). According to PC2, the samples 3, 4, 6 SW, 7 SC compared to 1, 2, 5 SW and to 8, 9, 10 SC rosehips showed differences at the seed region, mainly depending on the glucosidic bonds with the highest positive loadings at 312 and 473 cm$^{-1}$, in general owing to the C-C-O and C-C-C deformations of pectin. Another characteristic band associated with pectin is the lower intensity band at 856 cm$^{-1}$, which is considered a marker band of the α-glycosidic bonds in pectin [41]. Differently, samples 1, 2, 5 SW and 8, 9, 10 SC had characteristic signals at approximately 1600 cm$^{-1}$, suggesting a

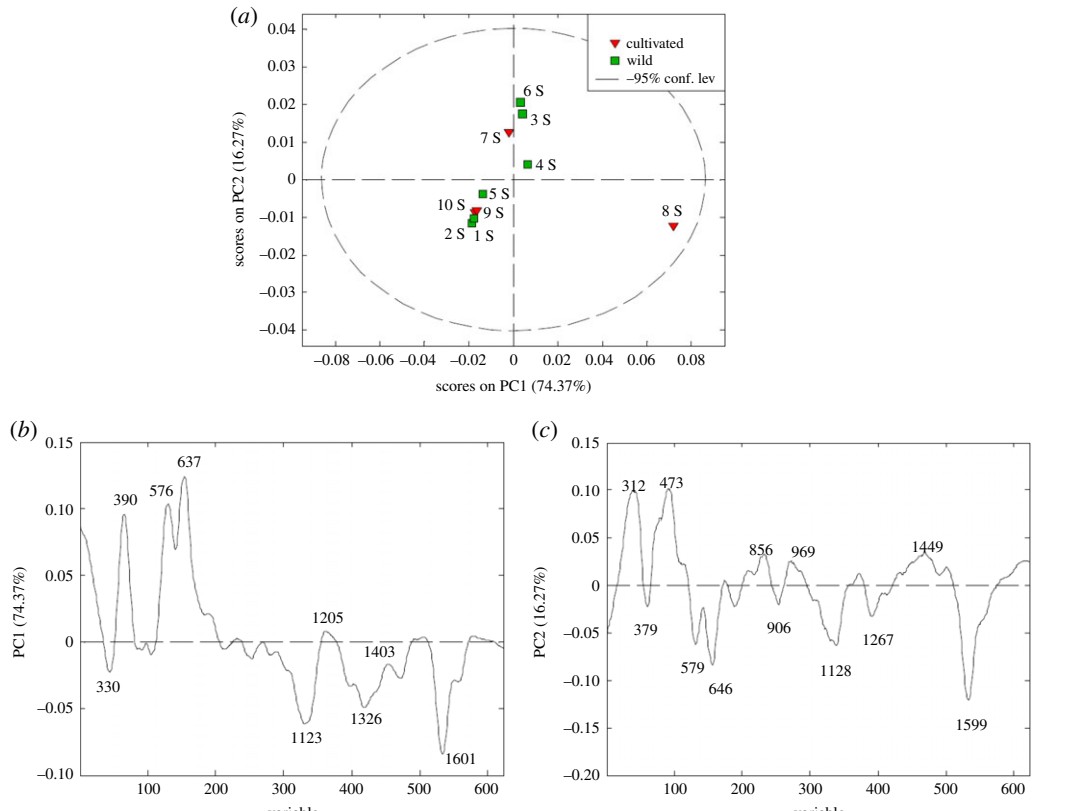

**Figure 5.** PCA analysis applied to the data obtained from Raman spectra of wild and cultivated rosehip seed samples: (*a*) score plot, (*b,c*) loading plots.

higher lignin contribution, probably owing to the C=C stretching vibrations of the aromatic phenolic compounds of the cell wall (e.g. *p*-coumaric acid) [25,43,59,61], a negative medium-intensity signal in the region from 570–650 cm$^{-1}$ assigned to bending of C–C–C, C–C–O of carbohydrates [48] and a lower intensity variable at approximately 1130 cm$^{-1}$, indicating the presence of coniferyl aldehyde [55,61,62]. The loading indicated the presence of C-H bending [25,37] which are involved in the unsaturation moieties of unsaturated fatty acids [63].

## 4. Conclusion

The results of the present study confirmed the potential of RS as a fast and sophisticated method for obtaining detailed information concerning the spatial distribution of plant metabolites in the studied rosehips. Although most assignments were performed only 'tentatively' or referred in general to the main chemical classes, it was shown that the rosehips examined are good sources of a wide range of phytonutrients, and the region below 1800 cm$^{-1}$ in Raman spectra was most significant for their characterization. The analysis of the Raman spectra related to hypanthium and seed, combined with the multivariate analysis (PCA) provided some differences in the chemical profiles of the rosehips grown in different Serbian habitats. Carotenoids, phenolic compounds and polysaccharides were the classes of compounds that contributed to the variation of hypanthium samples, whereas phenolic compounds, polysaccharides (pectin, cellulose and hemicellulose) and lipids (at a lower extent) were crucial for discriminating the seed samples. Notwithstanding, no clear differentiation between the cultivated and wild fruits with regard to both hypanthium and seed was observed. Further research will be focused on measuring the changes in carotenoids, phenolic compounds and fatty acids content that occur during the fruit development and maturation, using RS. The studied rosehip is a very interesting raw material for the development of new functional products, owing to the various applications of its bioactive compounds in gastronomy, food and cosmetic industries.

Data accessibility. This article has no additional data.

Authors' contributions. J.P.Dj. designed the study and provided plant material; I.P. performed Raman analysis; J.P.Dj. and I.P. analysed and interpreted spectroscopic data; Dj.K. performed statistical analyses; J.P.Dj., I.P., G.C. and Dj.K. discussed results, wrote and revised the manuscript. All authors approved the final version of the manuscript.

Competing interests. We declare we have no competing interests.

Funding. This study was supported by Faculty of Agriculture and Faculty of Chemistry in Belgrade and the Ministry of Education, Science, and Technological development of the Republic of Serbia within the agreements for scientific research work implemented in 2021, nos. 451-03-68/2020-14/200116 and 451-03-68/2020-14/200168.

Acknowledgements. The authors express their appreciation to Professor Luiz Fernando Cappa de Oliveira for his profound manuscript revision and numerous valuable recommendations.

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
