## [Peer Review File · Royal Society Open Science]

Review History

RSOS-202064.R0 (Original submission)

Review form: Reviewer 1

Is the manuscript scientifically sound in its present form?

No

Are the interpretations and conclusions justified by the results?

No

Is the language acceptable?

No

Do you have any ethical concerns with this paper?

No

Have you any concerns about statistical analyses in this paper?

No

Recommendation?

Major revision is needed (please make suggestions in comments)

Comments to the Author(s)

The manuscript describes the use of Raman spectroscopy to investigate and compare different samples of Rose hip, based on their chemical content. While the approach and subject matter are of interest, there are several issues which should be considered before publication.

- (i) The language and grammar should be checked throughout.
- (ii) The information on the value of Rosehip in the Republic of Serbia is more suited to the introduction than the summary. Is it valued in the rest of Europe? The world?
- (iii) The acronym PCA should be explained in the Summary, and all acronyms should be spelled out when first used in the text.
- (iv) "The absence of interference with water molecules...". Water does have a Raman scattering signal. It is, however relatively weak, although it is not absent!
- (v) "rosehips hypanthium parenchyma cells" are mentioned for the first time in Section 3.2. They should be mentioned earlier.
- (vi) "a frequency-doubled Nd/YAG laser.." normally has a wavelength of 532 nm (although in an Xplora system, a semiconductor diode laser of the same wavelength is usually employed). A 785 nm laser is also usually a semiconductor diode laser.
- (vii) How often was the instrument calibrated? Was it an "autocalibration" (I think Labspec 6) or a manual calibration (I think Labspec 5)?
- (viii) With reference to (vii), the shifts in spectra in Figure 2 and Figure 3 may be an indication that the calibration was different for different measurements. This is also supported by the loading of PC2 in Figure 6, which looks like a first derivative of PC1. It should be confirmed that the calibration for all measurements was the same.
- (ix) the loading of PV1 of Figure 6 is mainly dominated by a baseline drift below $\sim 950\text{cm}^{-1}$. apart from this baseline drift, all other identifiable spectral features are positive, and therefore do not identify an difference between samples.
- (x) the Intensity axis of Figure 5 should show numbers... where is the zero? This may who the origin of the baseline drift.
- (xi) All spectra, rather than just the means, should be input into the PCA plot, to show the intra- and inter sample variability.

Review form: Reviewer 2 (Uttam Pal)

Is the manuscript scientifically sound in its present form?

No

Are the interpretations and conclusions justified by the results?

No

Is the language acceptable?

Yes

Do you have any ethical concerns with this paper?

No

Have you any concerns about statistical analyses in this paper?

No

Recommendation?

Major revision is needed (please make suggestions in comments)

Comments to the Author(s)

In this article, authors reported that multivariate analysis of the fingerprint region of Raman spectra of the fresh ripe rosehip pulp can tell if the roship is a cultivar or a wild type. However, the similar analysis on the seeds could not distinguish a cultivar from the wild type. How the sample was prepared lacks clarity. Choice of different lasers to record the spectra for seed and pulp is not also clear. Many conclusions have been drawn that are not substantiated by the results. Specific comments are given below:

- 1) The title says "characterization of storage reserves", however, no temporal data is given. It is not even clear from the methods, how long the rosehips were stored and in which condition before the spectra were measured. It is important as the samples were collected from different geographical locations.
- 2) Regarding the preparation of the seeds two statements were made: a) "for Raman spectroscopy analysis, seeds were separated from hypanthium for each sample, and grinded to fine powder in a mill." b) "Raman microspectroscopy of rose hip fruit was focusing on direct measurement of rosehips hypanthium parenchyma cells and its seeds. The seeds were longitudinally cut at room temperature..." Please clarify how the seeds were prepared for Raman measurement.
- 3) Wouldn't it be better to record spectra from homogenized pulp?
- 4) Why 532 nm laser was used for the fleshy part and 785 nm for seeds? The fleshy part contains more light active molecules and fluorophores. Any fluorescence would suppress weak signals in Raman spectra. Wouldn't it be better to use 785 nm laser for the fleshy parts as well.
- 5) The authors claimed the method to be a "novel screening tool" developed by the authors, however, PCA coupled with Raman spectroscopy is already a widely used method.

Many conclusions have been drawn which are not substantiated by the results and purely hypothetical:

- 6) The authors conclude that the method could be used "...fast selection of rosehips...without fruit destruction." It would be relevant for sorting individual fruits even from the same harvest. However, the authors undertook no such study to show that the method can actually distinguish different quality fruits from the same harvest. Instead, they averaged over many samples from the same harvest.
- 7) The authors conclude that the method could "optimize the production of high-quality fruit." This study could only distinguish wild and cultivated varieties. From this study it cannot be concluded if the low and high quality fruit could be distinguished by this method. The seeds contained widely different levels of phytochemicals as evident from the Raman spectra of the seeds. However, PCA could not distinguish the seeds. It can be argued that the method shall fail to distinguish high and low quality fruits.
- 8) On "fruit development and maturation," as well, no study was undertaken. All the results are on mature fruits.

Additional experiments maybe performed:

9) FT-IR is complementary to Raman spectroscopy. It would be interesting to see if PCA of FT-IR spectra can lead to the similar results.

Minor comments:

10) Figure 3 and 5 may be moved to supporting information.

11) Authors may tabulate the peak positions of figure 3 to highlight the peak shift.

12) "The ancient Romans believed that...bite of rabid dogs." Should accompany a statement that no scientific association has been found.

13) "...used in folk medicine in wound-healing and skin disease treatments [5]." Should accompany a statement if there is any scientific proof and cite the primary sources.

Decision letter (RSOS-202064.R0)

This year has been very difficult for everyone, and we want to take the opportunity to thank you for your continued support in 2020.

The Royal Society Open Science editorial office will be closed from the evening of Friday 18 December 2020 until Monday 4 January 2021. We will not be responding during this time. If you have received a deadline within this time period, please contact us as soon as possible to allow us to extend the deadline. If you receive any automated messages during this time asking you to meet a deadline, we offer apologies and invite you to respond after the festive period or during normal working hours.

With our best for a peaceful festive period and New Year, and we look forward to working with you in 2021.

Dear Professor Popović-Djordjević:

Title: Rapid characterisation of storage reserves in wild and cultivated rosehip: application of Raman microscopy combined with multivariate analysis

Manuscript ID: RSOS-202064

The editor assigned to your manuscript has now received comments from reviewers. We would like you to revise your paper in accordance with the referee and Subject Editor suggestions which can be found below (not including confidential reports to the Editor). Please note this decision does not guarantee eventual acceptance.

Please submit your revised paper before 15-Jan-2021. Please note that the revision deadline will expire at 00.00am on this date. If we do not hear from you within this time then it will be assumed that the paper has been withdrawn. In exceptional circumstances, extensions may be possible if agreed with the Editorial Office in advance. We do not allow multiple rounds of revision so we urge you to make every effort to fully address all of the comments at this stage. If deemed necessary by the Editors, your manuscript will be sent back to one or more of the original reviewers for assessment. If the original reviewers are not available we may invite new reviewers.

To revise your manuscript, log into <http://mc.manuscriptcentral.com/rsos> and enter your Author Centre, where you will find your manuscript title listed under "Manuscripts with

Decisions." Under "Actions," click on "Create a Revision." Your manuscript number has been appended to denote a revision. Revise your manuscript and upload a new version through your Author Centre.

RSC Associate Editor:
Comments to the Author:
(There are no comments.)

RSC Subject Editor:
Comments to the Author:
(There are no comments.)

Reviewers' Comments to Author:
Reviewer: 1

Comments to the Author(s)

The manuscript describes the use of Raman spectroscopy to investigate and compare different samples of Rose hip, based on their chemical content. While the approach and subject matter are of interest, there are several issues which should be considered before publication.

- (i) The language and grammar should be checked throughout.
- (ii) The information on the value of Rosehip in the Republic of Serbia is more suited to the introduction than the summary. Is it valued in the rest of Europe? The world?
- (iii) The acronym PCA should be explained in the Summary, and all acronyms should be spelled out when first used in the text.
- (iv) "The absence of interference with water molecules...". Water does have a Raman scattering signal. It is, however relatively weak, although it is not absent!

- (v) "rosehips hypanthium parenchyma cells" are mentioned for the first time in Section 3.2. They should be mentioned earlier.
- (vi) "a frequency-doubled Nd/YAG laser.." normally has a wavelength of 532 nm (although in an Xplora system, a semiconductor diode laser of the same wavelength is usually employed). A 785 nm laser is also usually a semiconductor diode laser.
- (vii) How often was the instrument calibrated? Was it an "autocalibration" (I think Labspec 6) or a manual calibration (I think Labspec 5)?
- (viii) With reference to (vii), the shifts in spectra in Figure 2 and Figure 3 may be an indication that the calibration was different for different measurements. This is also supported by the loading of PC2 in Figure 6, which looks like a first derivative of PC1. It should be confirmed that the calibration for all measurements was the same.
- (ix) the loading of PV1 of Figure 6 is mainly dominated by a baseline drift below $\sim 950\text{cm}^{-1}$. apart from this baseline drift, all other identifiable spectral features are positive, and therefore do not identify an difference between samples.
- (x) the Intensity axis of Figure 5 should show numbers... where is the zero? This may who the origin of the baseline drift.
- (xi) All spectra, rather than just the means, should be input into the PCA plot, to show the intra- and inter sample variability.

Reviewer: 2

Comments to the Author(s)

In this article, authors reported that multivariate analysis of the fingerprint region of Raman spectra of the fresh ripe rosehip pulp can tell if the roship is a cultivar or a wild type. However, the similar analysis on the seeds could not distinguish a cultivar from the wild type. How the sample was prepared lacks clarity. Choice of different lasers to record the spectra for seed and pulp is not also clear. Many conclusions have been drawn that are not substantiated by the results. Specific comments are given below:

- 1) The title says "characterization of storage reserves", however, no temporal data is given. It is not even clear from the methods, how long the rosehips were stored and in which condition before the spectra were measured. It is important as the samples were collected from different geographical locations.
- 2) Regarding the preparation of the seeds two statements were made: a) "for Raman spectroscopy analysis, seeds were separated from hypanthium for each sample, and grinded to fine powder in a mill." b) "Raman microspectroscopy of rose hip fruit was focusing on direct measurement of rosehips hypanthium parenchyma cells and its seeds. The seeds were longitudinally cut at room temperature..." Please clarify how the seeds were prepared for Raman measurement.
- 3) Wouldn't it be better to record spectra from homogenized pulp?
- 4) Why 532 nm laser was used for the fleshy part and 785 nm for seeds? The fleshy part contains more light active molecules and fluorophores. Any fluorescence would suppress weak signals in Raman spectra. Wouldn't it be better to use 785 nm laser for the fleshy parts as well.
- 5) The authors claimed the method to be a "novel screening tool" developed by the authors, however, PCA coupled with Raman spectroscopy is already a widely used method.

Many conclusions have been drawn which are not substantiated by the results and purely hypothetical:

6) The authors conclude that the method could be used “...fast selection of rosehips...without fruit destruction.” It would be relevant for sorting individual fruits even from the same harvest. However, the authors undertook no such study to show that the method can actually distinguish different quality fruits from the same harvest. Instead, they averaged over many samples from the same harvest.

7) The authors conclude that the method could “optimize the production of high-quality fruit.” This study could only distinguish wild and cultivated varieties. From this study it cannot be concluded if the low and high quality fruit could be distinguished by this method. The seeds contained widely different levels of phytochemicals as evident from the Raman spectra of the seeds. However, PCA could not distinguish the seeds. It can be argued that the method shall fail to distinguish high and low quality fruits.

8) On “fruit development and maturation,” as well, no study was undertaken. All the results are on mature fruits.

Additional experiments maybe performed:

9) FT-IR is complementary to Raman spectroscopy. It would be interesting to see if PCA of FT-IR spectra can lead to the similar results.

Minor comments:

10) Figure 3 and 5 may be moved to supporting information.

11) Authors may tabulate the peak positions of figure 3 to highlight the peak shift.

12) “The ancient Romans believed that...bite of rabid dogs.” Should accompany a statement that no scientific association has been found.

13) “...used in folk medicine in wound-healing and skin disease treatments [5].” Should accompany a statement if there is any scientific proof and cite the primary sources.

Author's Response to Decision Letter for (RSOS-202064.R0)

See Appendix A.

RSOS-202064.R1 (Revision)

Review form: Reviewer 1

Is the manuscript scientifically sound in its present form?

Yes

Are the interpretations and conclusions justified by the results?

Yes

Is the language acceptable?

Yes

Do you have any ethical concerns with this paper?

No

Have you any concerns about statistical analyses in this paper?

Yes

Recommendation?

Accept with minor revision (please list in comments)

Comments to the Author(s)

Although the authors have addressed many of the issues raised during the initial review, there remain some concerns, particularly in relation to the PCA.

Figure 4 indicates that, those samples which score positively according to PC1 (a) have strong carotenoid features (b), and that those that score negatively have a strong background tail between (in (b) this is "Variable 0- 250"). There are no clearly identifiable chemical features which are associated with the negatively scoring samples in Fig 4(a). This suggests an issue of baselining the spectra.

Review form: Reviewer 2 (Uttam Pal)

Is the manuscript scientifically sound in its present form?

Yes

Are the interpretations and conclusions justified by the results?

Yes

Is the language acceptable?

Yes

Do you have any ethical concerns with this paper?

No

Have you any concerns about statistical analyses in this paper?

No

Recommendation?

Accept as is

Comments to the Author(s)

The authors adequately addressed all the issues in the revised manuscript. It is suitable for publication in its current form.

Decision letter (RSOS-202064.R1)

Dear Professor Popović-Djordjević:

Title: Rapid characterisation of hypanthium and seed in wild and cultivated rosehip: application of Raman microscopy combined with multivariate analysis
Manuscript ID: RSOS-202064.R1

Thank you for submitting the above manuscript to Royal Society Open Science. On behalf of the Editors and the Royal Society of Chemistry, I am pleased to inform you that your manuscript will be accepted for publication in Royal Society Open Science subject to minor revision in accordance with the referee suggestions. Please find the reviewers' comments at the end of this email.

The reviewers and handling editors have recommended publication, but also suggest some minor revisions to your manuscript. Therefore, I invite you to respond to the comments and revise your manuscript.

Because the schedule for publication is very tight, it is a condition of publication that you submit the revised version of your manuscript before 29-Jan-2021. Please note that the revision deadline will expire at 00.00am on this date. If you do not think you will be able to meet this date please let me know immediately.

Supplementary files will be published alongside the paper on the journal website and posted on the online figshare repository (<https://figshare.com>). The heading and legend provided for each supplementary file during the submission process will be used to create the figshare page, so

please ensure these are accurate and informative so that your files can be found in searches. Files on figshare will be made available approximately one week before the accompanying article so that the supplementary material can be attributed a unique DOI.

Kind regards,
Dr Laura Smith
Publishing Editor, Journals

RSC Associate Editor:
Comments to the Author:
(There are no comments.)

RSC Subject Editor:
Comments to the Author:
(There are no comments.)

Reviewer comments to Author:
Reviewer: 1

Comments to the Author(s)
Although the authors have addressed many of the issues raised during the initial review, there remain some concerns, particularly in relation to the PCA. Figure 4 indicates that, those samples which score positively according to PC1 (a) have strong carotenoid features (b), and that those that score negatively have a strong background tail between (in (b) this is "Variable 0- 250"). There are no clearly identifiable chemical features which are associated with the negatively scoring samples in Fig 4(a). This suggests an issue of baselining the spectra.

Reviewer: 2

Comments to the Author(s)
The authors adequately addressed all the issues in the revised manuscript. It is suitable for publication in its current form.

Author's Response to Decision Letter for (RSOS-202064.R1)

See Appendix B.

Decision letter (RSOS-202064.R2)

Dear Professor Popović-Djordjević:

Title: Rapid characterisation of hypanthium and seed in wild and cultivated rosehip: application of Raman microscopy combined with multivariate analysis
Manuscript ID: RSOS-202064.R2

It is a pleasure to accept your manuscript in its current form for publication in Royal Society Open Science. The chemistry content of Royal Society Open Science is published in collaboration with the Royal Society of Chemistry.

RSC Associate Editor
Comments to the Author:
(There are no comments.)

Reviewer(s)' Comments to Author:

Appendix A

Royal Society Open Science Editor-in-Chief

Dear editor,

Please find enclosed the revised version of the manuscript entitled,

"Rapid characterization of **hypanthium and seed** in wild and cultivated rosehip: application of Raman microscopy combined with multivariate analysis ", ID: **RSOS-202064**

I would to acknowledge you and the reviewers for dedicated work on our manuscript and a great contribution for its scientific and technical quality improvement. Accordingly, we accepted all suggestions given and replied to all comments.

Text has been English edited by a professional, and changes are written in red font.

Changes are highlighted in yellow in the manuscript file and Supplementary material.

Please note that we have slightly changed the manuscript title (highlighted in yellow) as in the present form is more in line with reviewers' comments.

I hope that you and reviewers will approve this change.

In the following lines you may find detailed responses to the reviewers' comments.

Sincerely yours,

Jelena Popović-Djordjević, PhD
University of Belgrade
Faculty of Agriculture
Department for Food technology and Biochemistry
Belgrade
Serbia
jelenadj@agrif.bg.ac.rs

Reviewers' Comments to Author:

Reviewer: 1

Comments to the Author(s)

The manuscript describes the use of Raman spectroscopy to investigate and compare different samples of Rose hip, based on their chemical content. While the approach and subject matter are of interest, there are several issues which should be considered before publication.

(i) The language and grammar should be checked throughout.

Authors: The English has been checked by a professional.

(ii) The information on the value of Rosehip in the Republic of Serbia is more suited to the introduction than the summary. Is it valued in the rest of Europe? The world?

Authors: Thank you for the valuable comment. The sentence has been moved into Introduction in adopted form.

(iii) The acronym PCA should be explained in the Summary, and all acronyms should be spelled out when first used in the text.

Authors: Full name of PCA was added in summary. Also, all abbreviations are explained when first mentioned.

(iv) "The absence of interference with water molecules...". Water does have a Raman scattering signal. It is, however relatively weak, although it is not absent!

Authors: We appreciate this comment. Text is re-phrased in revised version.

(v) "rosehips hypanthium parenchyma cells" are mentioned for the first time in Section 3.2. They should be mentioned earlier.

Authors: We have mentioned it in Summary as suggested.

(vi) "a frequency-doubled Nd/YAG laser.." normally has a wavelength of 532 nm (although in an Xplora system, a semiconductor diode laser of the same wavelength is usually employed). A 785 nm laser is also usually a semiconductor diode laser.

Authors: Corrections have been made in the text.

(vii) How often was the instrument calibrated? Was it an "autocalibration" (I think Labspec 6) or a manual calibration (I think Labspec 5)?

Authors: Yes, it was auto calibrated each time before recording of spectra. The text is corrected according to the comment.

(viii) With reference to (vii), the shifts in spectra in Figure 2 and Figure 3 may be an indication that the calibration was different for different measurements. This is also supported by the loading of

PC2 in Figure 6, which looks like a first derivative of PC1. It should be confirmed that the calibration for all measurements was the same.

Authors: In figures 2 and 3 the small shifts are a consequence of heterogeneous plant material and there are regularly present interactions of carotenoids with other cell constituents (Schulz et al., 2006; Baranska et al., 2006). The shifts are not-connected with the already done calibration. All Raman shifts below 3cm^{-1} (i.e. below used resolution) cannot be considered for interpretation since they are below resolution.

Autocalibration (i.e. procedure include check all lasers and gratings using a silicone as reference) was done every time before starting the process of spectral recording for all laser and all available gratings in combination with objectives.

As we pointed out above, calibration procedure was performed frequently and in accordance with manufacturer recommendation. We used basic spectra for PCA analysis (without transformation into derivative form). Sharp peaks in loadings plots are results of relatively high resolution used in our measurements as well as the good initial spectra.

Schulz, H., Schütze, W. & Baranska, M. (2006). Fast determination of carotenoids in tomatoes and tomato products by Raman spectroscopy. *Acta Horticulture* 712, 901–906. DOI: <https://doi.org/10.17660/ActaHortic.2006.712.118>

Baranska, M, Schultze, W & Schulz, H. (2006). Determination of Lycopene and β -Carotene Content in Tomato Fruits and Related Products: Comparison of FT-Raman, ATR-IR, and NIR, Spectroscopy. *Analytical Chemistry*, 78, 8456–8461. DOI: <https://doi.org/10.1021/ac061220j>

(ix) the loading of PV1 of Figure 6 is mainly dominated by a baseline drift below $\sim 950\text{cm}^{-1}$. apart from this baseline drift, all other identifiable spectral features are positive, and therefore do not identify an difference between samples.

Authors: Variables with the highest positive contribution along PC1 are dominant in samples at the right side of the plot (1, 2HW and 7-9 HC). Consequently, these variables are responsible for differentiation of samples along PC1 axis, despite the fact that there are no variables with a negative contribution. Also, signals above 950cm^{-1} (in the region from ~ 1000 to 1530cm^{-1}) are expected for hypanthium samples because carotenoids are the dominant compounds in these samples.

(x) the Intensity axis of Figure 5 should show numbers... where is the zero? This may who the origin of the baseline drift.

Authors: We have presented raw spectra of all hypanthium and seed individual samples in the revised supporting material document. Intensity numbers are included in those spectra.

(xi) All spectra, rather than just the means, should be input into the PCA plot, to show the intra- and inter sample variability.

Authors: Raman spectra of analyzed samples were recorded as repeated measurements, but we used mean values as representative of each sample, in order to examine the similarities and differences between the samples. The main goal of this paper was the characterization of hypanthium and seed samples. While, intra-sample variability requires a different approach to data processing, and this was not the subject of our research.

Reviewer: 2

Comments to the Author(s)

In this article, authors reported that multivariate analysis of the fingerprint region of Raman spectra of the fresh ripe rosehip pulp can tell if the roship is a cultivar or a wild type. However, the similar analysis on the seeds could not distinguish a cultivar from the wild type. How the sample was prepared lacks clarity. Choice of different lasers to record the spectra for seed and pulp is not also clear. Many conclusions have been drawn that are not substantiated by the results. Specific comments are given below:

Authors: We appreciate your valuable insight into the manuscript.

1) The title says “characterization of storage reserves”, however, no temporal data is given. It is not even clear from the methods, how long the rosehips were stored and in which condition before the spectra were measured. It is important as the samples were collected from different geographical locations.

Authors: Thank you for this observation. Accordingly, we clarified samples preparation procedure in M&M section.

2) Regarding the preparation of the seeds two statements were made: a) “for Raman spectroscopy analysis, seeds were separated from hypanthium for each sample, and grinded to fine powder in a mill.” b) “Raman microspectroscopy of rose hip fruit was focusing on direct measurement of rosehips hypanthium parenchyma cells and its seeds. The seeds were longitudinally cut at room temperature...” Please clarify how the seeds were prepared for Raman measurement.

Authors: Samples preparation has been described in M&M, subsection 3.1.

3) Wouldn't it be better to record spectra from homogenized pulp?

Authors: Yes of course. We clarified it in M&M, subsection 3.1.

4) Why 532 nm laser was used for the fleshy part and 785 nm for seeds? The fleshy part contains more light active molecules and fluorophores. Any fluorescence would suppress weak signals in Raman spectra. Wouldn't it be better to use 785 nm laser for the fleshy parts as well.

Authors: Our previous investigation on fresh fruit organs (Pećinar, 2019, Kolašinac et al. 2018) and preliminary investigation on rosehips indicated the intensive signals from carotenoids using 532 nm laser before we start collecting the samples for the current analysis. Considering to the high Raman activity of the carotenoid molecules and the resonance excitation in the fluorescence-free, longer wavelength spectral range at 532 nm was used for the sensitive detection of the molecule's highly specific Raman response (Zeise et al., 2018). Using this positive practice in spectra recording for the same type of material we use 532 nm laser for analysis.

Raman spectroscopy has been used extensively for fatty acid detection in vegetable oil samples (Dong et al., 2013; Huang et al., 2016) and in our previous study (Alimpić Aradski et al. 2020) good results in *Salvia* sp. seed composition at 785 nm were obtained.

Pećinar, I. (2019): Raman Microscopy in Plant Science, Carotenoids Detection in Fruit Material. In: Vucelić Radović, B., Lazić, D. and Nikšić, M. (eds.) Application of Molecular Methods and Raman Microscopy/Spectroscopy in Agricultural Sciences and Food Technology, Pp. 177–186. London: Ubiquity Press. DOI: <https://doi.org/10.5334/bbj.n>. License: CC-BY 4.0

Kolašinac, S., Pećinar, I., Lević, S., Rančić, D., Dajić Stevanović, Z., Schulz, H. (2018): Raman spectroscopic characterization of carotenoids from rose hips herbal tea mixtures, 10th CMAPSEEC, Book of abstracts, May 20-24, Split, Croatia, pp 121.

Zeise, I., Heiner, Z., Holz, S., Joester, M., Büttner, C. & Kneipp, J. (2018). Raman Imaging of Plant Cell Walls in Sections of *Cucumis sativus*, *Plants* 7, 7, DOI: <https://doi.org/10.3390/plants7010007>

Dong, W., Zhang, Y., Zhang, B., & Wang, X. (2013). Rapid prediction of fatty acid composition of vegetable oil by Raman spectroscopy coupled with least squares support vector machines. *Journal of Raman Spectroscopy*, 44(12), 1739-1745.

Huang, F., Li, Y., Guo, H., Xu, J., Chen, Z., Zhang, J., & Wang, Y. (2016). Identification of waste cooking oil and vegetable oil via Raman spectroscopy. *Journal of Raman Spectroscopy*, 47(7), 860-864.

Alimpić Aradski, A., Janošević, D., Pećinar, I., Budimir, S., Dajić Stevanović, Z., Matevski, V., Marin P.D. & Duletić-Laušević, S. (2020): Micromorphological and anatomical characteristics of *Salvia amplexicaulis* Lam., *S. jurisicii* Košanin and *S. ringens* Sibth. & Sm. (Lamiaceae), *Plant Biosystems - An International Journal Dealing with all Aspects of Plant Biology*, DOI: 10.1080/11263504.2020.1727976

5) The authors claimed the method to be a “novel screening tool” developed by the authors, however, PCA coupled with Raman spectroscopy is already a widely used method.

Authors: This is a misunderstanding... We did not certainly develop the method. The application of the method for screening phytochemical composition of rosehip is a new approach. No literature data was found except the paper of Prof. de Oliveira (2008).

de Oliveira VE, Castro HV, Edwards HGM, de Oliveira LFC. (2010) Carotenes and carotenoids in natural biological samples: a Raman spectroscopic analysis. *J. Raman Spectrosc.* 41, 642–650. <https://doi.org/10.1002/jrs.2493>

Many conclusions have been drawn which are not substantiated by the results and purely hypothetical:

6) The authors conclude that the method could be used “...fast selection of rosehips...without fruit destruction.” It would be relevant for sorting individual fruits even from the same harvest. However, the authors undertook no such study to show that the method can actually distinguish different quality fruits from the same harvest. Instead, they averaged over many samples from the same harvest.

7) The authors conclude that the method could “optimize the production of high-quality fruit.” This study could only distinguish wild and cultivated varieties. From this study it cannot be concluded if the low and high quality fruit could be distinguished by this method. The seeds contained widely different levels of phytochemicals as evident from the Raman spectra of the seeds. However, PCA could not distinguish the seeds. It can be argued that the method shall fail to distinguish high and low quality fruits.

8) On “fruit development and maturation,” as well, no study was undertaken. All the results are on mature fruits.

Authors: Thank you for this valuable comment. According to comments 6, 7 and 8, we have re-written the Conclusion section. Some “claims” were more consideration of our research in the future.

Additional experiments maybe performed:

9) FT-IR is complementary to Raman spectroscopy. It would be interesting to see if PCA of FT-IR spectra can lead to the similar results.

Authors: We absolutely agree with this observation. Unfortunately, in this situation and due to Covid-19, our Faculty is closed until 11th of January and we are not able to perform additional analyses. But FT-IR will be considered in future research.

In addition, the FT-IR technique could be useful for seed material in a case on fatty acids rich plant organs (Czekus et al., 2019; Alimpić Aradski et al. 2020) but not for carotenoid rich fruit as shown in our publication (Pećinar, 2019) and other our reports and preliminary analysis. From that point of view, we have chosen Raman instrumentation for spectral analysis of both fruit compartments.

Czekus, B., Pećinar, I., Petrović, I., Paunović, N., Savić, S., Jovanović, Z., Stikić, R. (2019): Raman and Fourier transform infrared spectroscopy application to the *Puno* and *Titicaca* cvs. of quinoa seed microstructure and perisperm characterization, *Journal of Cereal Science*, 87, 25-30.

Alimpić Aradski, A., Janošević, D., Pećinar, I., Budimir, S., Dajić Stevanović, Z., Matevski, V., Marin P.D., Duletić-Laušević, S. (2020): Micromorphological and anatomical characteristics of *Salvia amplexicaulis* Lam., *S. jurisicii* Košanin and *S. ringens* Sibth. & Sm. (Lamiaceae), *Plant Biosystems - An International Journal Dealing with all Aspects of Plant Biology*, 155, 92-108. DOI: 10.1080/11263504.2020.1727976

Pećinar, I. (2019): Raman Microscopy in Plant Science, Carotenoids Detection in Fruit Material. In: Vucelić Radović, B., Lazić, D. and Nikšić, M. (eds.) *Application of Molecular Methods and Raman Microscopy/Spectroscopy in Agricultural Sciences and Food Technology*, Pp. 177–186. London: Ubiquity Press. <https://doi.org/10.5334/bbj.n>

Minor comments:

10) Figure 3 and 5 may be moved to supporting information.

Authors: Figures 3 and 5 were removed from the main text. Figures of raw spectra for all individual samples are included in supporting material. Accordingly, relevant corrections have been made in the main text.

11) Authors may tabulate the peak positions of figure 3 to highlight the peak shift.

Authors: The Raman shifts are already mentioned in the text

12) “The ancient Romans believed that...bite of rabid dogs.” Should accompany a statement that no scientific association has been found.

Authors: The sentence has been amended as suggested.

13) “...used in folk medicine in wound-healing and skin disease treatments [5].” Should accompany a statement if there is any scientific proof and cite the primary sources.

Authors: Correction has been made in the text and the relevant reference was cited.

Appendix B

Royal Society Open Science Editor-in-Chief

Dear editor,

I would to acknowledge you and the reviewers for dedicated work on our manuscript. Accordingly, we accepted suggestions given and replied to the comments.

Changes are highlighted in yellow in the manuscript file and Supplementary material.

In the following lines you may find detailed responses to the reviewers' comments.

Sincerely yours,

Jelena Popović-Djordjević, PhD
University of Belgrade
Faculty of Agriculture
Department for Food technology and Biochemistry
Belgrade, Serbia
jelenadj@agrif.bg.ac.rs

Responses to Reviewers comments

Comments to the Author(s)

Reviewer: 1

Although the authors have addressed many of the issues raised during the initial review, there remain some concerns, particularly in relation to the PCA.

Figure 4 indicates that, those samples which score positively according to PC1 (a) have strong carotenoid features (b), and that those that score negatively have a strong background tail between (in (b) this is "Variable 0- 250"). There are no clearly identifiable chemical features which are associated with the negatively scoring samples in Fig 4(a). This suggests an issue of baselining the spectra.

Authors: Thank you for this valuable observation. We have noticed our mistake in baselining the spectra for hypanthium samples. So, we have done the baseline correction again, and obtained results are presented in the re-revised version of manuscript. Section 4.3. is re-written accordingly.

References are renumbered in the main text, reference list and supporting material, as the new one [27] is added in the section 3.3. (M&M).

Reviewer: 2

Comments to the Author(s)

The authors adequately addressed all the issues in the revised manuscript. It is suitable for publication in its current form.

Authors: Thank you for the positive comment on the revised manuscript